# Optimizing a human monoclonal antibody for better neutralization of SARS-CoV-2

Qian Wang[1,2,3,9], Yicheng Guo [1,3,9], Ryan G. Casner[4,5,7,9], Jian Yu[1], Manoj S. Nair [1], Jerren Ho[1], Eswar R. Reddem[4,5], Ian A. Mellis[1], Madeline Wu[1], Chih-Chen Tzang [1], Hsiang Hong [1], Yaoxing Huang [1,2,3], Lawrence Shapiro[1,4,5] ✉, Lihong Liu[1,3,8] ✉ & David D. Ho [1,2,3,6] ✉

SARS-CoV-2 has largely evolved to resist antibody pressure, with each successive viral variant becoming more and more resistant to serum antibodies in the population. This evolution renders all previously authorized anti-spike therapeutic monoclonal antibodies inactive, and it threatens the remaining pipelines against COVID-19. We report herein the isolation of a human monoclonal antibody with a broad but incomplete SARS-CoV-2 neutralization profile, but structural analyses and mutational scanning lead to the engineering of variants that result in greater antibody flexibility while binding to the viral spike. Three such optimized monoclonal antibodies neutralize all SARS-CoV-2 strains tested with much improved potency and breadth, including against subvariants XEC and LP.8.1. The findings of this study not only present antibody candidates for clinical development against COVID-19, but also introduce an engineering approach to improve antibody activity via increasing conformational flexibility.

Since the emergence of COVID-19 in late 2019, its causative agent, SARS-CoV-2, has undergone considerable evolution driven by immune pressure, with each successive viral variant becoming more resistant to serum antibodies elicited by prior infection and/or vaccination. By late 2020, mutations in the viral spike protein resulted in a discernible antigenic drift that yielded the Alpha variant, which evaded antibodies targeting the antigenic supersite on the N-terminal domain (NTD)[1–3]. Shortly thereafter, further antigenic drift led to the Beta and Gamma variants that evaded not only NTD-directed antibodies but also some antibodies directed to so-called class 1 and class 2 regions of the receptor-binding domain (RBD)[1,4,5]. These variants were, in turn, rapidly replaced by the Delta variant that dominated much of 2021,

and it evaded antibodies directed to select class 2 and class 3 epitopes on the RBD[6]. Importantly, this initial wave of SARS-CoV-2 variants knocked out three therapeutic monoclonal antibodies (mAbs) authorized to treat COVID-19 infection—etesevimab, bamlanivimab, and casirvimab[6–10].

The first major SARS-CoV-2 antigenic shift occurred with the emergence of the Omicron (BA.1) variant in late 2021 in southern Africa[11,12], likely due to a saltatory event resulting in an accumulation of 34 spike mutations. This variant rapidly gained dominance in the population, as a consequence of its exceptional resistance to serum antibodies in the population[13–18]. Specifically, mutations in Omicron impaired the neutralizing activity of antibodies targeting all regions

[1]Aaron Diamond AIDS Research Center, Columbia University Vagelos College of Physicians and Surgeons, New York, NY, USA. [2]Pandemic Research Alliance unit at the Wu Center for Pandemic Research, Columbia University Vagelos College of Physicians and Surgeons, New York, NY, USA. [3]Division of Infectious Diseases, Department of Medicine, Columbia University Vagelos College of Physicians and Surgeons, New York, NY, USA. [4]Zuckerman Mind Brain Behavior Institute, Columbia University, New York, NY, USA. [5]Department of Biochemistry and Molecular Biophysics, Columbia University Vagelos College of Physicians and Surgeons, New York, NY, USA. [6]Department of Microbiology and Immunology, Columbia University Vagelos College of Physicians and Surgeons, New York, NY, USA. [7]Present address: Large Molecules Research, Sanofi R&D, Cambridge, MA, USA. [8]Present address: State Key Laboratory of Virology and Taikang Center for Life and Medical Sciences, Wuhan University, Wuhan, Hubei, China. [9]These authors contributed equally: Qian Wang, Yicheng Guo, Ryan G. Casner. ✉e-mail: lss8@cumc.columbia.edu; ll3411@cumc.columbia.edu; dh2994@cumc.columbia.edu

(classes 1–4) of the RBD, as well as two additional clinically authorized mAbs—imdevimab and tixagevimab[9,13,19]. Antigenic drift from BA.1 then ensued successively throughout 2022 and yielded subvariants BA.2, BA.5, and BQ.1.1, each being more antibody resistant than its predecessor. This evolutionary drift also knocked out the utility of three more clinically authorized mAbs— sotrovimab, ciligavimab, and bebtelovimab[19–23].

The second major antigenic shift was detected in late 2022 when a recombination event between BA.2 and BA.2.75 resulted in the XBB subvariant, which promptly evolved into XBB.1.5, EG.5.1, HK.3, and HV.1 via further antigenic drift[11,24–26]. Again, each progeny subvariant was more antibody resistant than its predecessor, largely by evading more antibodies directed to the class 1 region of RBD. Sequentially, these subvariants dominated much of 2023 until the third major antigenic shift led to the emergence of BA.2.86, which presumably was due to another saltatory event, again in southern Africa, leading to an accumulation of >30 mutations in the BA.2 spike[27]. This new subvariant was again highly resistant to serum neutralizing antibodies, including those targeting the subdomain 1 (SD1) region of spike[28–30]. With one additional spike mutation, L455S, BA.2.86 became JN.1, and the JN.1 sublineage has been dominant worldwide from late 2023 until recently[31,32].

In March 2024, a monoclonal antibody known as pemivibart (VYD222) was authorized for clinical use as pre-exposure prophylaxis for immunocompromised individuals who do not respond robustly to COVID-19 vaccines[33]. This antibody was specifically engineered to have broad activity against many known SARS-CoV-2 strains, including JN.1. However, as the JN.1 sublineage continued its antigenic drift, new forms such as XEC began to replace their predecessor in recent months. As of April 2025, the dominant SARS-CoV-2 strains are XEC and LP.8.1[34]. Alarmingly, these subvariants are already ~18–28-fold more resistant to pemivibart in vitro than JN.1[35,36]. This looming threat to the only authorized pre-exposure prophylaxis for millions of immunocompromised individuals who live in fear of COVID-19 is a stark reminder of the necessity to continue to develop more medical interventions to keep up with SARS-CoV-2 evolution. In addition, another mAb, AstraZeneca's sipavibart (AZD3152), is also in clinical trials; however, this antibody has been shown to be affected by the F456L mutation[37], which is present in multiple dominant JN.1 sublineages, such as XEC and LP.8.1.

We now describe the isolation and characterization of a human monoclonal antibody, 19-77, with a broad but incomplete SARS-CoV-2 neutralization profile. Structural analyses and mutational scanning led to the construction of engineered antibody variants, 19-77Δ, with a single amino acid substitution at residue R71 in the framework region 3 of the heavy chain. These modifications resulted in greater flexibility of the complementarity-determining regions (CDRs) of the heavy chain while binding to the viral spike, thereby enabling three 19-77Δ variants to neutralize all SARS-CoV-2 strains tested with much improved potency and breadth, including against the JN.1 sublineages. The findings of this study not only offer new antibody candidates for pre-exposure prophylaxis against COVID-19 for immunocompromised individuals, but also introduce a unique engineering approach to improve antibody activity.

## Results

### Isolation and characterization of mAb 19-77

To isolate new neutralizing mAbs against SARS-CoV-2, we studied Donor 19 who had been infected by the Omicron BA.5 subvariant despite having received three doses of the original monovalent vaccine (BNT162b2) and one dose of the WT/BA.5 bivalent vaccine (mRNA-1273.222) (Supplementary Fig. 1a). His serum, obtained ~7 months after breakthrough infection, strongly neutralized the D614G strain but less potently against XBB.1.5 and SARS-CoV (Supplementary Fig. 1b). Given the dominance of XBB.1.5 at the start of this study, XBB.1.5 and

SARS-CoV spikes were used as probes to sort antigen-specific memory B cells from his peripheral blood mononuclear cells (PBMCs). The proportions of his antigen-specific B cells were 0.16% for XBB.1.5 spike, 0.067% for SARS-CoV, and 0.12% for both XBB.1.5 and SARS-CoV spikes; in contrast, PBMCs from two healthy donors showed negligible antigen-specific B cells (Supplementary Fig. 1c).

Donor 19's memory B cells from quadrants 2 and 3 (Supplementary Fig. 1c) were then subjected to single-cell RNA sequencing using 10X Genomics to obtain paired heavy and light chain sequences of each B cell receptor, as previously described[38]. A total of 113 paired sequences were selected, and the corresponding antibodies were synthesized for in vitro characterization. A majority (75) of the mAbs bound the D614G spike at a concentration of 10 μg/ml, but only 29 neutralized either D614G or EG.5.1 in vitro (Supplementary Fig. 1d). Among the neutralizing mAbs, 19-77 was chosen for further studies, because it not only bound strongly to the D614G spike and RBD by immunoassays (Fig. 1a), but also showed broadly neutralizing activity against many SARS-CoV-2 strains (Fig. 1b). Specifically, in pseudovirus assays, 19-77 neutralized D614G, Alpha, Beta, Gamma, Delta, BA.1, BA.2, BA.5, BQ.1.1, XBB.1.5, EG.5.1, BA.2.86, and JN.1 with $IC_{50}$ values < 0.05 μg/ml. However, its neutralizing activity was lower against HK.3 and JF.1, and not detectable against JD.1.1 or SARS-CoV.

Genetic analysis revealed that the heavy chain of 19-77 utilized IGHV3-53 (Supplementary Fig. 1e), showing that this antibody belonged to the well-studied VH3-53/66 class of antibodies targeting the SARS-CoV-2 spike. Its light chain originated from IGKV3-11. Notably, the CDRH3 of 19-77 contained only 9 amino acids, one of the shortest among published VH3-53/66 class mAbs (Fig. 1c). Moreover, the level of somatic hypermutation of its heavy chain was the highest observed among VH3-53/66 mAbs, and its light chain was also quite hypermutated (Fig. 1c). These striking genetic features demonstrated the uniqueness of 19-77 and prompted further studies.

To understand the molecular basis for the neutralization breadth and potency of 19-77, we then visualized its Fab fragment in complex with the D614G spike (Fig. 1d, Supplementary Fig. 2, and Supplementary Fig. 3a, b) by single-particle cryo-electron microscopy (cryo-EM). The Fab was predominantly bound to the spike protein in a three-RBD-up conformation, like other VH3-53/66 class antibodies targeting the class 1 region[39]. The primary interaction between the 19-77 heavy chain and the RBD was mediated by CDRH1 and CDRH2, with hydrogen bonds to A475, N487, Y473, D420, and Y421 (Fig. 1e). The CDRH3 region formed two hydrogen bonds with the RBD: one between R97 and the backbone carbonyl of N487, and another between E102 and Y489. On the other hand, the 19-77 light chain made contact with the apical ridge of the RBD via its CDRL1 and CDRL3 (Fig. 1f).

### Structural explanation for the loss of 19-77 activity against certain Omicron subvariants

The antibody footprint of 19-77 on the RBD largely overlapped with the human ACE2 receptor-binding site (Fig. 2a). Sequence conservation analysis indicated that most regions of its epitope are relatively conserved, except for the upper half of the left shoulder, which exhibits significant sequence variations at residues 455, 456, 475, 478, and 486. These specific residues have undergone frequent mutations in certain recent subvariants, such as EG.5.1, HK.3, JF.1, and JD.1.1. It is therefore not surprising that the neutralizing activity of 19-77 against HK.3 (XBB.1.9.2 carrying Q52H and L455F/F456L mutations) and JF.1 (XBB.1.16.6 carrying E180V, L455F/F456L, and K478R) decreased by 14- and 42-fold, respectively, compared with that against XBB.1.5 (Fig. 1b). The inactivity of 19-77 against JD.1.1 is likely due to its A475V mutation that causes a steric clash with CDRH1 (Fig. 2b). To further understand the resistance mechanism of L455F/F456L mutations to the neutralization by 19-77, we determined the structure of 19-77 Fab in complex with the RBD of EG.5.1 by X-ray crystallography, as well as the cryo-EM structures of the D614G RBD and the HK.3 RBD in complex

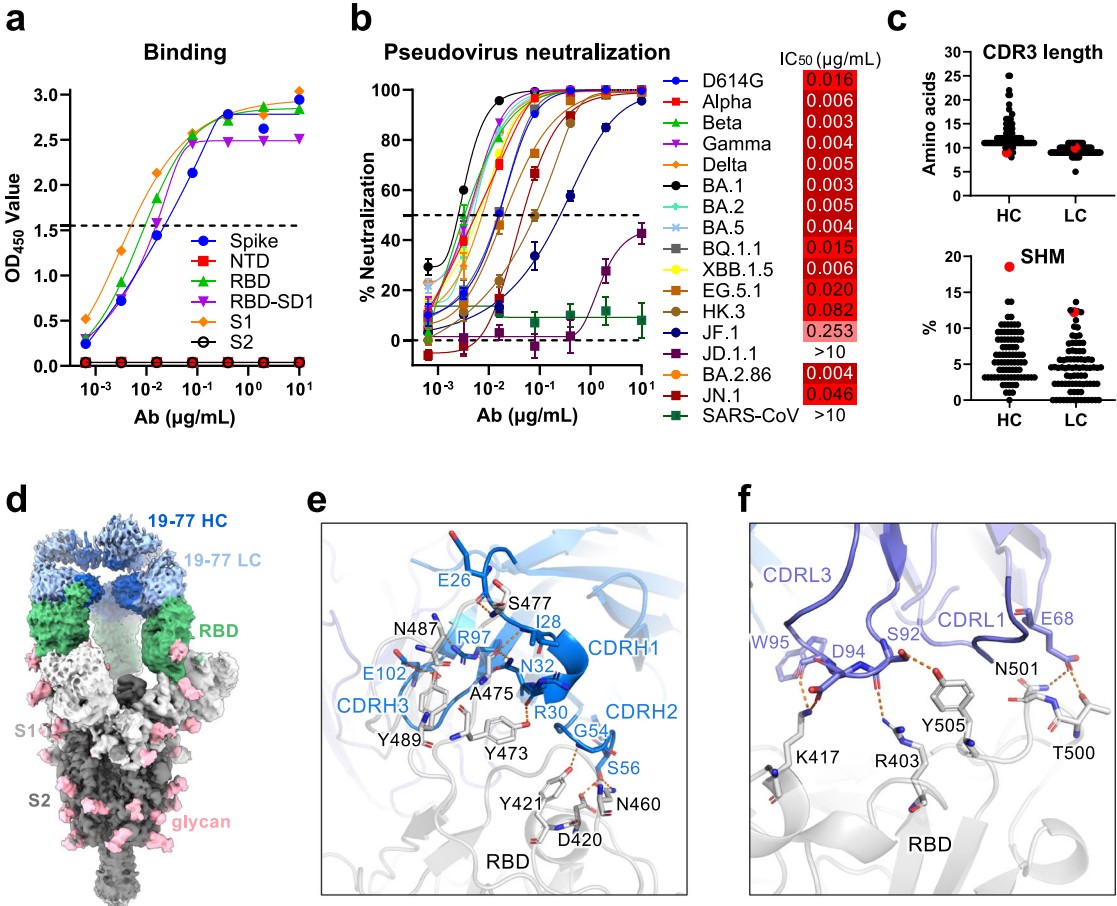

**Fig. 1 | Neutralization activity and epitope mapping of 19-77. a** Binding of 19-77 to the indicated antigens of SARS-CoV-2 D614G tested by ELISA. **b** Neutralization IC$_{50}$ values of 19-77 against pseudotyped SARS-CoV-2 variants and SARS-CoV, with the IC$_{50}$ of each virus is denoted. Neutralization curves are shown as mean ± standard error of mean (SEM) from technical triplicates. **c** Complementarity-determining region 3 (CDR3) lengths and somatic hypermutations (SHM) of 19-77 heavy chain (HC) and light chain (LC), compared with other 3-53 and 3-66 antibodies. 19-77 is highlighted in red. **d** Cryo-EM reconstruction of 19-77 in complex with the SARS-CoV-2 D614G spike protein, at a resolution of 2.45 Å. **e** Key interactions between 19-77 heavy chain and the D614G RBD. Hydrogen bonds are shown as the orange dashed lines. **f** Key interactions between 19-77 light chain and the D614G RBD. Orange dashed lines indicate hydrogen bonds.

with the Fabs of 19-77 and S309 (sotrovimab) (Supplementary Fig. 3), a non-overlapping mAb used to increase the molecular mass for visualization by cryo-EM. Compared with the D614G RBD (Fig. 2c), the change from phenylalanine (F) to leucine (L) at residue 456 in the EG.5.1 RBD reduced hydrophobic interactions with P100$_{HC}$ in the CDRH3 of 19-77 (Fig. 2d). Additionally, the L455F mutation in HK.3 introduced a minor van der Waals clash with P100$_{HC}$ (Fig. 2e).

### Optimization of 19-77

Given the minor to moderate steric clashes between the RBDs of resistant viruses and 19-77, we explored the possibility of optimizing the antibody to accommodate the resistance mutations. Utilizing in silico modeling and the energy-calculation algorithm FoldX[40], we performed saturation mutagenesis on each residue of the heavy chain, comparing the binding energy changes (ΔΔG) between each mutant and the original 19-77 in complex with the HK.3 RBD that contained the L455F/F456L mutations. Both beneficial and detrimental mutations were identified (Supplementary Fig. 4a). Interestingly, substitutions at residue R71 in framework region 3 of the heavy chain consistently predicted a moderately beneficial effect (-0.5 kcal/mol) on the binding of 19-77 to the HK.3 RBD. We then conducted alanine scanning on the antigen-contact residues of both heavy and light chains of 19-77, in addition to the residue R71, followed by testing each modified antibody for neutralization against JD.1.1, HK.3, EG.5, and EG.5-A475V (Fig. 3a, Supplementary Fig. 4b, and Supplementary Fig. 4c). Only the

R71A mutation significantly enhanced the neutralization activity of 19-77 against the viruses tested, including the restoration of neutralizing activity against JD.1.1 and EG.5-A475V. We therefore introduced the other 18 possible amino acid substitutions to the R71 position of 19-77 and then evaluated their neutralization efficacy against the same panel of resistant viruses (Supplementary Fig. 4d). Except for R71K, all other substitutions resulted in not only improved neutralization activities against EG.5 and HK.3 relative to the original antibody, but also restoration of activity against JD.1.1 and EG.5-A475V. Overall, the extent of improvement for these mutant antibodies (19-77Δs) ranged from 3- to >100-fold, with aliphatic substitutions R71A, R71L, and R71V being the best (Fig. 3b and Supplementary Fig. 4d).

Structural analysis showed that R71 of 19-77 forms hydrogen bonds with S30 of CDRH1 and P53 of CDRH2, likely contributing to antibody stability (Fig. 3c). Additional modeling indicated that mutating R71 could disrupt these hydrogen bonds, with the exception of R71K, which could also form similar interactions (Fig. 3d). We then hypothesized that R71 might restrict the motion of CDRH1 and CDRH2 of 19-77, and that mutating this arginine could lead to greater antibody flexibility. To test this hypothesis, we conducted molecular dynamics (MD) simulations for 19-77 with either arginine or alanine at position 71 of the heavy chain. The results showed that A71 increased the range of motion of the heavy-chain CDRs compared to R71 (Supplementary Movie 1). Moreover, the fluctuations in the movement of each residue for the A71 variant of 19-77 were noticeably greater than those in the

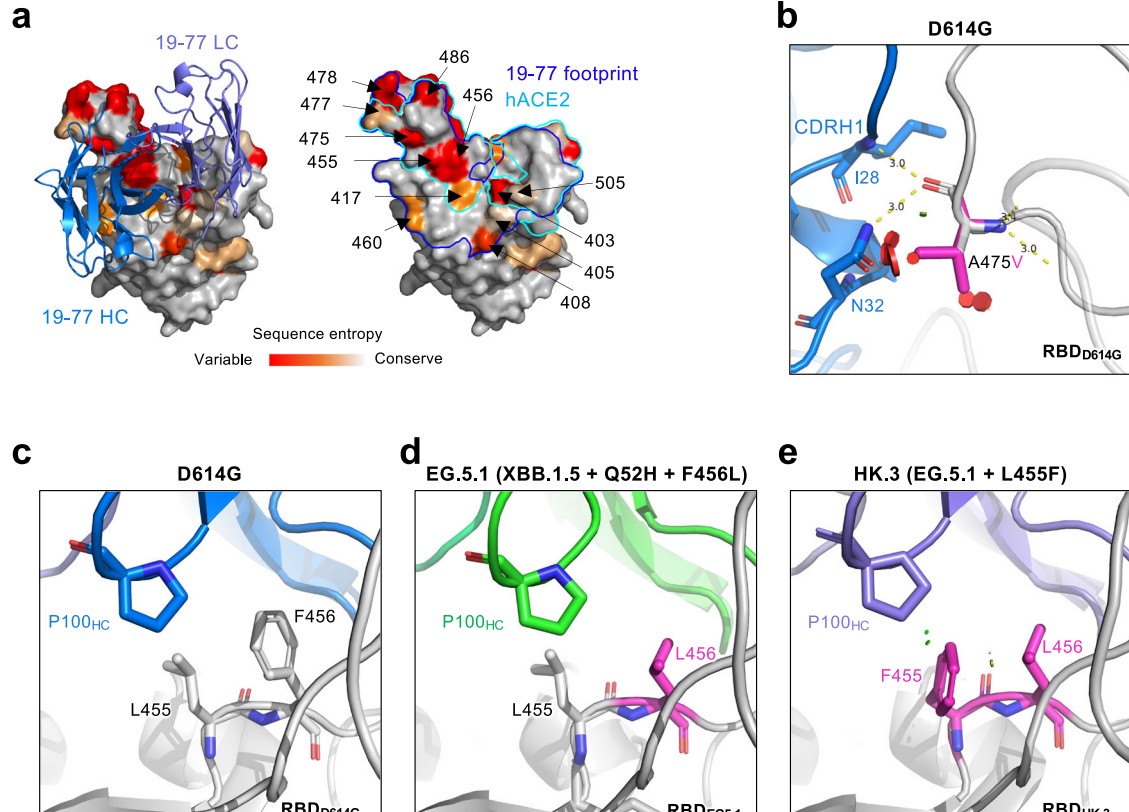

**Fig. 2 | Sequence conservation and resistant mutations in the 19-77 epitope.** **a** Top view of the RBD inner face in complex with the 19-77 antibody. 19-77 heavy chain and light chain are shown in marine and light blue, respectively. The residues in the RBD are colored by the sequence entropy in circulating SARS-CoV-2 variants. The blue and cyan boundaries show the footprints of 19-77 and human ACE2

(hACE2), respectively. **b** Structure modeling of how A475V on the RBD affects 19-77 neutralization. The clashes are shown as red plates. **c**–**e** Comparison of residues 455 and 456 on RBD and P100HC in D614G (**c**), EG.5.1 (**d**), and HK.3 (**e**) structures. The van der Waals clashes are shown as green plates.

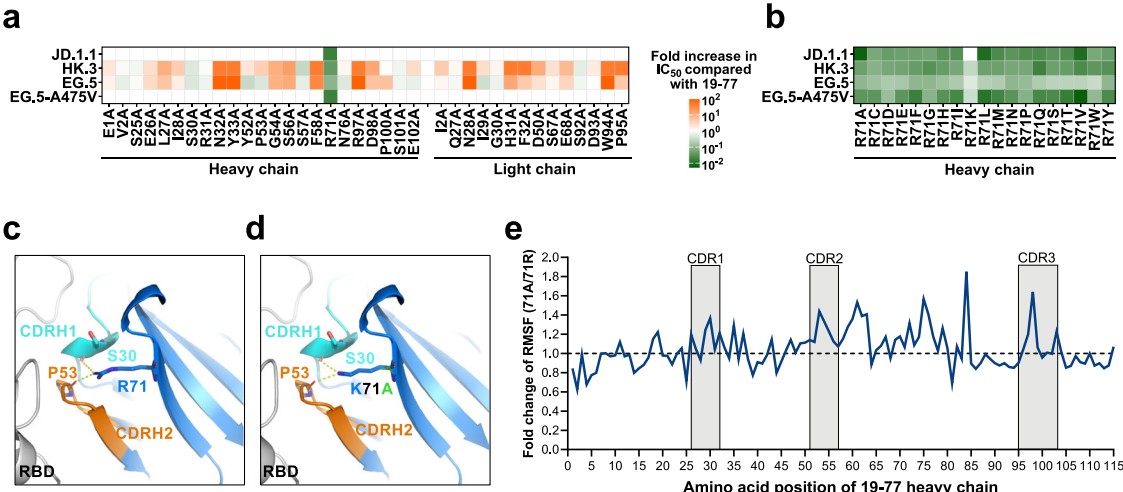

**Fig. 3 | Optimization of 19-77 in neutralization activity. a** Neutralization of 19-77 carrying individual mutations on both heavy and light chains, in comparison with the wild-type 19-77 antibody. **b** Neutralization of 19-77 carrying various amino acid substitutions at R71 on the heavy chain, compared with the wild-type 19-77 antibody. **c** Depiction of the interactions between R71 and the heavy chain CDRs H1 (cyan) and H2 (orange) in 19-77. Dashed lines indicate the presence of hydrogen

bonds. **d** Structure modeling of K71 (blue) and A71 (green) with HCDRs of 19-77. Dashed lines indicate the presence of hydrogen bonds between 19-77ΔK. **e** Comparison of the RMSF between the Fab regions of 19-77 and its mutant 19-77ΔA. The Y-axis represents the fold change in RMSF per-residue in 19-77ΔA relative to 19-77. The X-axis displays the positional numbering of residues within the antibody. RMSF, per-residue root-mean-square fluctuation.

| Clade | Pseudovirus | 19-77 | 19-77 mutants | | | Control Abs | |
|---|---|---|---|---|---|---|---|
| | | | R71A | R71L | R71V | P4J15 | VYD222 |
| SARS-CoV-2 variants | D614G | 0.015 | 0.009 | 0.007 | 0.009 | 0.004 | 0.013 |
| | B.1.1.7 (Alpha) | 0.003 | 0.003 | 0.003 | 0.003 | 0.003 | 0.012 |
| | B.1.351 (Beta) | 0.003 | 0.003 | 0.002 | 0.002 | 0.003 | 0.017 |
| | P.1 (Gamma) | 0.002 | 0.002 | 0.002 | 0.002 | 0.003 | 0.028 |
| | B.1.617.2 (Delta) | 0.005 | 0.004 | 0.004 | 0.004 | 0.003 | 0.011 |
| | BA.1 | 0.001 | 0.001 | 0.001 | 0.001 | 0.001 | 0.203 |
| | BA.2 | 0.008 | 0.004 | 0.004 | 0.004 | 0.003 | 0.102 |
| | BA.5 | 0.006 | 0.004 | 0.004 | 0.004 | 0.010 | 0.101 |
| | BA.2.75 | 0.015 | 0.007 | 0.007 | 0.009 | 0.004 | 0.516 |
| | CH.1.1 | 0.010 | 0.006 | 0.009 | 0.009 | 0.026 | 0.381 |
| | DV.7.1 | 0.285 | 0.019 | 0.031 | 0.025 | 1.179 | 1.250 |
| | XBC.1.6 | 0.007 | 0.003 | 0.005 | 0.004 | 0.033 | 0.183 |
| | BQ.1.1 | 0.011 | 0.008 | 0.006 | 0.009 | 0.023 | 0.260 |
| | XBB.1.5 | 0.008 | 0.005 | 0.004 | 0.008 | 0.059 | 0.215 |
| | XBB.1.16.6 | 0.054 | 0.004 | 0.002 | 0.003 | 0.017 | 0.243 |
| | XBB.2.3 | 0.012 | 0.007 | 0.008 | 0.009 | 0.018 | 0.263 |
| | EG.5.1 | 0.020 | 0.008 | 0.006 | 0.008 | 0.039 | 0.233 |
| | FL.1.5.1 | 0.056 | 0.006 | 0.006 | 0.008 | 0.039 | 0.240 |
| | JF.1 | 0.156 | 0.008 | 0.006 | 0.008 | 0.227 | 0.343 |
| | HV.1 | 0.052 | 0.006 | 0.005 | 0.006 | 0.020 | 0.159 |
| | HK.3 | 0.107 | 0.007 | 0.006 | 0.008 | 1.340 | 0.304 |
| | JD.1.1 | >10 | 0.033 | 0.032 | 0.058 | 0.355 | 0.288 |
| | BA.2.87.1 | 0.026 | 0.008 | 0.009 | 0.009 | 0.003 | 1.043 |
| | BA.2.86 | 0.006 | 0.003 | 0.003 | 0.003 | 0.081 | 0.459 |
| | JN.1 | 0.046 | 0.005 | 0.005 | 0.006 | >10 | 0.070 |
| | JN.4 | 0.017 | 0.004 | 0.003 | 0.003 | 0.270 | 0.512 |
| | LB.1 | 0.087 | 0.008 | 0.009 | 0.008 | 2.303 | 0.208 |
| | KP.2 | 0.031 | 0.004 | 0.005 | 0.004 | >10 | 0.148 |
| | KP.3 | 0.585 | 0.044 | 0.072 | 0.068 | 0.729 | 0.554 |
| | KP.2.3 | 0.107 | 0.015 | 0.009 | 0.005 | >10 | 0.459 |
| | KP.3.1.1 | 1.426 | 0.066 | 0.088 | 0.075 | 2.862 | 1.217 |
| | XEC | >10 | 0.092 | 0.200 | 0.068 | >10 | 1.998 |
| | LP.8.1 | >10 | 0.105 | 0.175 | 0.072 | >10 | 0.939 |
| | LP.8.1.1 | >10 | 0.095 | 0.114 | 0.058 | >10 | 0.845 |
| | MC.10.1 | >10 | 0.324 | 0.278 | 0.083 | >10 | >10 |
| | LF.7 | 0.26 | 0.02 | 0.017 | 0.012 | >10 | 0.405 |
| SARS-CoV-2 like sarbecoviruses | GX-pangolin | 1.878 | 0.302 | 0.317 | 0.163 | >10 | 0.008 |
| | RaTG13 | 0.005 | 0.003 | 0.002 | 0.003 | 0.004 | >10 |
| | BANAL52 | 0.027 | 0.015 | 0.007 | 0.011 | 0.007 | 0.008 |
| | BANAL236 | 0.012 | 0.007 | 0.005 | 0.011 | 0.003 | 0.005 |
| | GD-pangolin | 0.009 | 0.006 | 0.004 | 0.006 | 0.003 | 0.003 |

IC$_{50}$ (µg/mL)  | <0.01 | 0.01-0.1 | 0.1-1 | 1-10 | >10 |

**Fig. 4 | Neutralization activity of 19-77 and 19-77ΔA/L/V against the indicated pseudoviruses.** Neutralization IC$_{50}$ values of the indicated antibodies against a panel of pseudotyped viruses, including SARS-CoV-2 variants and SARS-CoV-2-like sarbecoviruses.

original antibody, especially within the CDRs (Fig. 3e). These findings, in turn, suggested that the increased flexibility of the A71 variant could allow the antibody to better accommodate for certain mutations in the SARS-CoV-2 spike.

We next assessed the in vitro neutralizing activity of the R71A, R71L, and R71V versions of 19-77 (designated 19-77ΔA, 19-77ΔL, and 19-77ΔV, respectively) against a large pseudovirus panel comprising 36 SARS-CoV-2 variants and 5 SARS-CoV-2-like sarbecoviruses. Control antibodies included the original 19-77, along with the recently authorized pemivibart (VYD222)[33] and a published broadly neutralizing mAb, P4J15[41] (Fig. 4). The engineered 19-77 variants consistently showed better virus-neutralizing potency and breadth compared to the parental antibody. Importantly, against viruses that are widely circulating today, the improvements were substantial: 7.7-to-9.2-fold for JN.1, 6.2-to-7.8-fold for KP.2, 8.1-to-13.3-fold for KP.3, 16.2-to-21.6-fold for KP.3.1.1, and >50.0-to-147.1-fold for XEC, with the latter two

being most relevant presently. The activity P4J15 was either severely impaired or knocked out by JN.1 and its progeny viruses. Pemivibart showed decreased potency against recent JN.1 sublineages, particularly KP.3.1.1 and XEC, and completely lost activity against MC.10.1. These results showed that the optimized 19-77 mAbs possess the requisite antiviral properties to be considered as candidates for clinical development, pending a full developability assessment. Reassuringly, the three mutations at R71 did not adversely impact antibody expression, aggregation propensity, or pharmacokinetics in mice (Supplementary Fig. 5).

**Impact of R71 mutations on other VH3-53/66 class of mAbs to the viral spike**

Monoclonal 19-77 belonged to the VH3-53/66, multi-donor class of SARS-CoV-2-neutralizing mAbs that target the class 1 region of RBD[39]. We therefore explored whether the same set of R71 mutations could

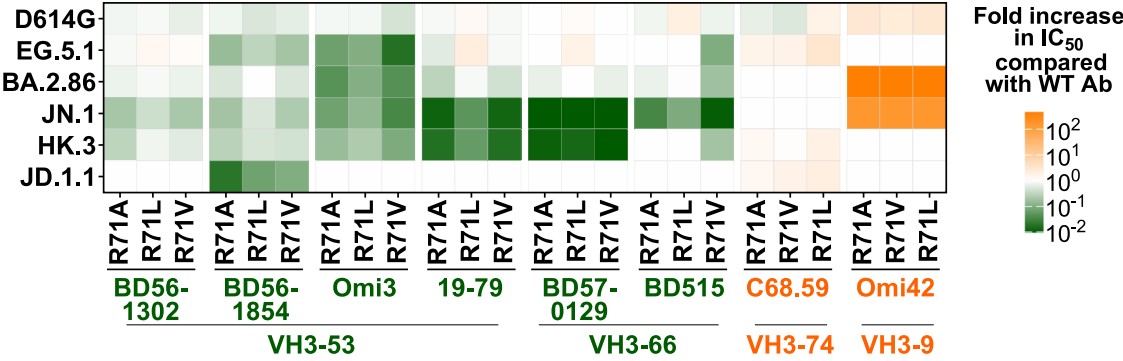

**Fig. 5 | R71A/L/V sensitizes the neutralization activity of other VH3-53/66 class antibodies.** Neutralization of the indicated antibodies from VH3-53/66 and other classes carrying R71A/L/V mutations, compared with the respective wild-type (WT) antibodies.

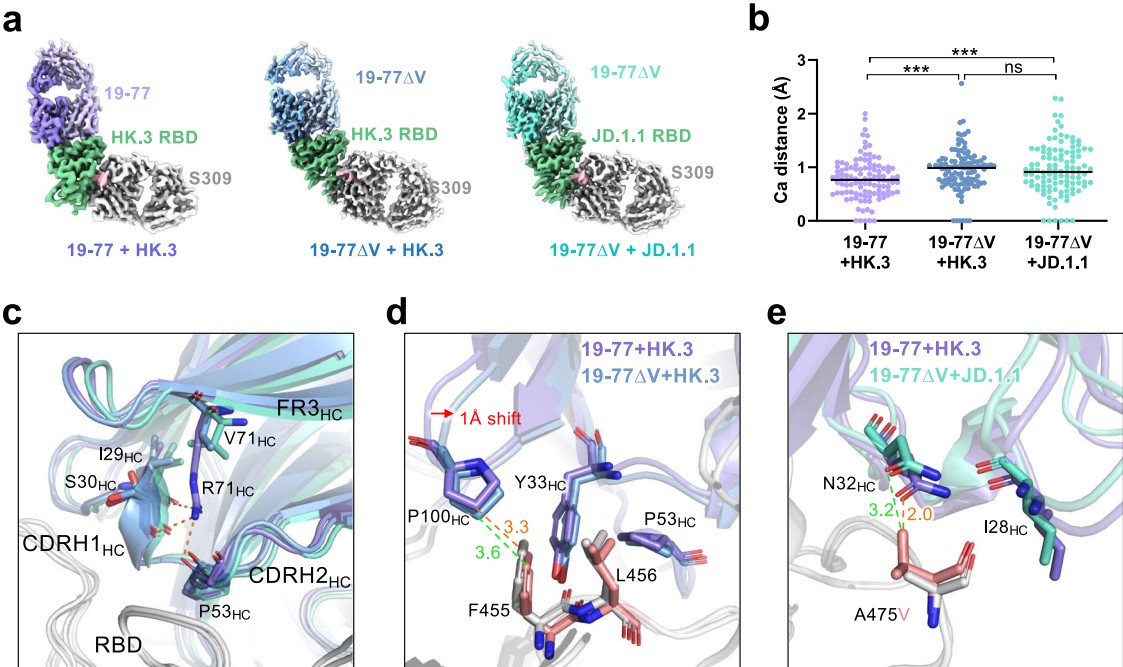

**Fig. 6 | Molecular basis of 19-77ΔV accommodating mutations in HK.3 and JD.1.1. a** Cryo-EM maps of 19-77 and 19-77ΔV Fabs bound to SARS-CoV-2 RBD variants. **b** Distribution of Cα distance for the heavy chain residues of 19-77 in **a** compared with 19-77 in the D614G complex. The structures are aligned based on RBD. Comparisons were made by student's t-tests. ***p < 0.001. ns not significant. **c** Comparison of CDRH1, CDRH2, and R71V in the alignments of 19-77 structures.

The hydrogen bonds are shown as orange dashed lines. **d** Interaction details of P100_{HC} in 19-77 and 19-77ΔV with F455 and L456 in the HK.3 RBD. The orange and green dashed lines represent the distances between P100_{HC} and F455 in 19-77 and 19-77ΔV, respectively. **e** Interaction details of N32_{HC} in 19-77 and 19-77ΔV with A475V in the D614G and JD.1.1 RBDs. The orange and green dashed lines represent the distances between V475 and N32 in 19-77 and 19-77ΔV, respectively.

similarly benefit other members of this antibody class. Mutations R71A, R71L, and R71V were then introduced into five VH3-53/66 class mAbs, including BD56-1302[42], BD56-1854[42], Omi-3[43], 19-79, BD57-0129[42], and BD-515[44], as well as two non-VH3-53/66 class mAbs, C68.59[29], and Omi42[43], serving as controls. The original mAbs and their modified antibody variants were then assayed for their neutralizing activity against a panel of pseudoviruses (Supplementary Fig. 6). All unmodified mAbs showed partial or complete loss of neutralizing activity against some later Omicron subvariants such as EG.5.1, HK.3, JD.1.1, and JN.1. In contrast, every modified VH3-53/66 mAb exhibited improved neutralization capability, albeit with varying magnitudes (3- to >100-fold), whereas no improvement was observed for non-VH3-53/66 mAbs (Fig. 5 and Supplementary Fig. 6). In fact, a number of modified mAbs regained the ability to neutralize subvariants that had been completely resistant to their unmodified counterparts. For example, BD57-0129 failed to neutralize JN.1 or HK.3, but its modified

counterparts neutralized both viruses robustly. The extension of the benefit of R71 mutations to mAbs other than 19-77 suggests that the responsible mechanism is not likely to operate solely at the interface with the paratope. Instead, the broader benefit observed is consistent with the hypothesis of greater conformational flexibility of the antibody CDRs discussed previously. Additionally, antibodies using VH3-53 in the human B cell repertoire very rarely carry a mutation at R71 (Supplementary Fig. 7), highlighting the uniqueness of our engineered mAbs compared to those found after SARS-CoV-2 infections.

**Molecular insights into enhanced virus neutralization by 19-77ΔV.** To further explore the mechanism of the enhancement of virus neutralization by mutating R71, we determined two additional cryo-EM structures: the Fabs of 19-77ΔV and S309 in complex with HK.3 or JD.1.1 RBD at 2.9 and 3.1 Å resolution, respectively (Fig. 6a and Supplementary Fig. 3). We then compared these

**a**

| mAb | Authentic virus for escape selection | Selection passages | Escape mutations | Frequency of mutations in GISAID | Relative infectivity of pseudotyped escape variant | Fold change in IC$_{50}$ of pseudotyped escape variants compared with the relative WT virus | | |
|---|---|---|---|---|---|---|---|---|
| | | | | | | 19-77 | 19-77ΔV | hACE2 |
| 19-77ΔV | JN.1 | P5 | F456S+K554E | 0.0002% | 64% | <-99 | -33 | -2.2 |

**b**

| | Escape mutations under sequentially increased 19-77ΔV concentrations | | | Properties of escape variants from 50 µg/ml of 19-77ΔV selection | | | | | |
|---|---|---|---|---|---|---|---|---|---|
| No. | 4 µg/ml | 20 µg/ml | 50 µg/ml | Frequency in GISAID | Relative infectivity | Fold change in IC$_{50}$ compared with JN.1 | | | |
| | | | | | | 19-77ΔA | 19-77ΔL | 19-77ΔV | hACE2 |
| 1 | T393A | T393A/Y489H | T393A/Y489H | 0% | 1% | <-1214 | <-1097 | <-1236 | <-114 |
| 2 | F456S | F456S | F456S/E471G | 0% | 2% | <-1214 | <-1097 | <-1236 | -103 |
| 3 | Y473S | Y473S | Y473S | 0.00036% | 1% | <-1214 | <-1097 | <-1236 | <-114 |
| 4 | A475D | A475D | A475D | 0.0035% | 2% | <-1214 | <-1097 | <-1236 | <-114 |
| 5 | G476D | G476D | G476D/N487H | 0% | 1% | <-1214 | <-1097 | <-1236 | <-114 |
| 6 | G485D | G485D | G485D | 0.010% | 33% | -9.0 | -12 | -10 | -12 |
| 7 | Y489H | Y489H | Y489H | 0.0028% | 18% | -44 | -57 | -121 | -27 |
| 8 | Y489H | Y489H | N405D/Y489H | 0% | 22% | <-1214 | <-1097 | <-1236 | -34 |

| Resistant | <-1000 | <-100 | <-10 | <-3 |
|---|---|---|---|---|

**c**

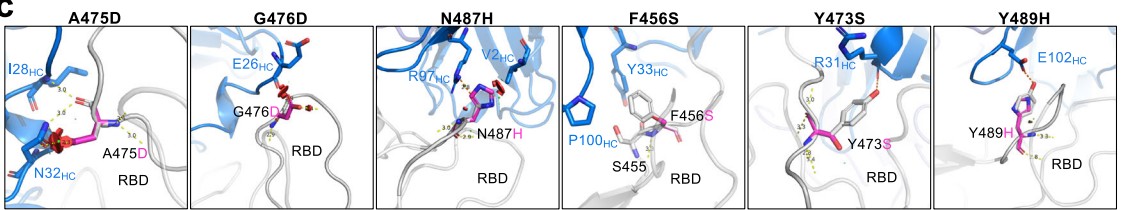

**Fig. 7 | In vitro selection of 19-77 resistant mutations. a** Properties of the authentic JN.1 escape variant selected under the pressure of 19-77ΔV. **b** Properties of the replication-competent VSV-JN.1 escape variants selected under the pressure of 19-77ΔV. **c** Structural modeling of how A475D, G476D, N487H, F456S, Y473S, and Y489H affect 19-77 neutralization. The clashes are shown as red plates, and the hydrogen bonds are shown as orange dashed lines.

structures by aligning their RBDs with that of the 19-77-D614G complex and measuring the alpha carbon (Cα) distances for each heavy chain residue in 19-77 against the D614G complex. The analysis showed that the Cα distances in 19-77ΔV, when bound to HK.3 and JD.1.1 RBDs, were significantly greater (0.99 and 0.91 Å, respectively) compared to the original 19-77 in complex with HK.3 RBD (0.76 Å) (Fig. 6b). Moreover, the motion of the CDRH1 and CDRH2 were observed directly across these structures, and the valine substitution at residue 71 resulted in the loss the hydrogen bonds that had existed between R71 and the residues S30 and P53 in the heavy chain (Fig. 6c). In addition, the CDRH3 in 19-77ΔV exhibited a shift of 1 Å compared to the original 19-77, resulting in an increased distance (from 3.3 to 3.6 Å) between residues P100 and F455 (Fig. 6d). This change mitigated the slight van der Waals clash between these two residues in HK.3 RBD and 19-77 complex (Fig. 2e). Lastly, the distance between V475 in the JD.1.1 RBD and N32 of the heavy chain of 19-77ΔV increased to 3.2 Å from the 2.0 Å found in the original 19-77 bound to D614G RBD (Fig. 6e), showing the mitigation of steric hindrance caused by the A475V mutation. Collectively, these findings support the notion that 19-77Δ mAbs are indeed more flexible and therefore more accommodating for the mutations found in emerging SARS-CoV-2 variants.

## In vitro selection of SARS

**-CoV-2 resistant to 19-77ΔV**. Although the SARS-CoV-2-neutralization breadth and potency of the optimized 19-77 variants were impressive (Fig. 4), our expectation was that the virus will still find a way to escape given its history of evading all COVID-19 mAbs authorized to date. We therefore undertook two sets of in vitro studies to define its mutational pathways to resist 19-77ΔV. First, we performed serial passaging of the authentic JN.1 virus in increasing antibody concentrations (0.1 to 50 µg/mL) in Vero-ACE2-TMPRSS2 cells over 15 days (under 3 days per passage). After five passages, a resistant virus was sequenced and found to carry mutations F456S (TTT to TCT) and K554E (AAG to GAG, Fig. 7a). When these mutations were introduced into the JN.1 pseudovirus, the resultant pseudovirus showed a 33-fold greater resistance

to 19-77ΔV, and even more to 19-77. Additional studies showed that the antibody evasion was attributable solely to F456S, while K554E partially compensated for the loss in viral fitness (Supplementary Fig. 8a and Supplementary Fig. 8b). Notably, this combination of mutations is extremely rare (0.0002%) in the GISAID database.

Second, we also utilized a replication-competent VSV bearing the JN.1 spike to select for variants that could escape from 19-77ΔV. This selection was performed in 66 replicates in 24-well plates, involving four rounds of selection over 8 days in Vero-E6-TMPRSS2-T2A-ACEs cells with increasing concentrations of 19-77ΔV (0.4 to 50 µg/mL). At the end of this process, escape variants were detected in 11 wells, and sequencing of their spike genes revealed eight distinct single or double mutations (Fig. 7b). Notably, all these mutations resulted from single nucleotide changes but were rarely found in the GISAID database, with frequencies ranging from 0 to 0.01%, and each of the pseudoviruses constructed with these mutations exhibited impaired infectivity compared to the JN.1 pseudovirus, ranging from 1 to 33% (Fig. 7b and Supplementary Fig. 8c). Furthermore, these pseudoviruses demonstrated increased resistance to soluble hACE2 inhibition (Fig. 7b and Supplementary Fig. 8d), indicating a loss in receptor affinity, perhaps accounting in part for their reduced infectivity. Single mutations G485D (GGC to GAC) and Y489H (TAC to CAC), both residing in the RBD, decreased the neutralization activity of 19-77ΔA/L/V considerably (9- to 121-fold), whereas the remaining mutations completely abolished the neutralization activity of the three optimized mAbs (Fig. 8b and Supplementary Fig. 8d).

In silico structural analysis showed that each of the escape viruses contained a mutation within the epitope of 19-77ΔV that impaired antibody binding (Fig. 7c). A475D, G476D, and N487H caused steric hindrance to antibody binding, the F465S mutation and residue S455 in JN.1 significantly reduced hydrophobic interactions between the antibody and RBD. Furthermore, both Y473S and Y489H abolished hydrogen bonds with the antibody, contributing to the dramatic drop in neutralization observed for the escape viruses with these mutations.

## Discussion

In this study, we present findings on the characterization of a human IGHV3-53-derived monoclonal antibody, 19-77, which exhibited neutralizing activity against most but not all SARS-CoV-2 variants (Figs. 1b, 4). Strikingly, 19-77 featured the shortest CDRH3 (nine amino acids) while containing the highest degree of somatic hypermutations among its class of mAbs (Fig. 1c). Structural analysis revealed that 19-77 recognized the spike RBD in the "up" position (Fig. 1d), and its epitope overlapped with the ACE2 footprint (Fig. 2a). Subsequent in silico energy calculations (Supplementary Fig. 4a) and empirical mutagenesis experiments (Figs. 3a, 4b) yielded modified antibodies 19-77ΔA, 19-77ΔL, and 19-77ΔV, each of which neutralized all 36 SARS-CoV-2 variants tested, including KP.3.1.1, XEC and LP.8.1.1 (Fig. 4). These optimized antibodies joined a limited list of human mAbs that possess sufficiently potent neutralizing activity against all viral variants, such as SA55[45], BD55-1205[46], and VIR-7229[47] that target an epitope overlapping with that of 19-77, as well as CYFN1006-1[48] that targets a RBD class 3 epitope. However, we showed that SARS-CoV-2 could easily escape from neutralization by 19-77ΔV in vitro (Fig. 7). In fact, the virus found multiple solutions to evade our optimized antibody, albeit with a fitness cost. Such an outcome was, more or less, expected given how SARS-CoV-2 has evolved in the population to render inactive all clinical mAbs authorized prior to 2024. Indeed, pemivibart (VYD222) that was authorized only months ago for use as prophylaxis against COVID-19 in immune deficient persons is already seriously threatened by the emergence of KP.3.1.1 and XEC subvariants[35,49].

It has been a daunting challenge, as well as a frustrating endeavor, to develop therapeutic or prophylactic mAbs to keep up with the rapid pace of SARS-CoV-2 evolution. Yet the reality is that millions of individuals worldwide are sufficiently immunocompromised that they cannot benefit from the protection conferred by COVID-19 vaccines. Such persons need effective prophylaxis with passively administered virus-neutralizing mAbs, as was conferred previously by the combination of tixagevimab and cilgavimab known as Evusheld[50] and presently by pemivibart[33]. Going forward, we believe one of our optimized 19-77Δ mAbs would qualify as a compelling candidate for clinical development given its superior potency and breadth against all SARS-CoV-2 variants known to date (Fig. 4), as well as its lack of discernible developability challenges to date (Supplementary Fig. 5). The potential clinical utility of an optimized 19-77Δ could be further improved if used in combination with a non-competing, broadly neutralizing mAb like CYFN1006-1[48]. Additionally, the use of antibody cocktails may help reduce the likelihood of generating escape variants during antiviral mAb therapy[51,52].

Herein, we also present a unique strategy for optimizing the activity of a monoclonal antibody by increasing its conformational flexibility. Traditionally, antibody engineering had been focused primarily on increasing the binding affinity using a number of approaches, including chain shuffling[53,54], site-directed mutagenesis[55], phage and other display technologies[56–60], and structure-based or artificial-intelligence-guided methods[61–63]. Such methodologies, typically, had been designed to increase antibody affinity by modifying the CDR residues at the antibody-antigen interface. Other antibody approaches to optimize antibody activity included the use of multivalency with the creation of "multabodies"[64], or the strategic addition of a glycan to bulk up a steric hindrance effect[65]. The framework regions of a mAb were seldom touched, because they are crucial for maintaining the proper orientation and conformation of the CDRs[66,67]. In our study on 19-77, however, in silico energy calculations indicated that a single amino acid substitution at residue 71 in framework region 3 of the heavy chain could improve antibody binding to the viral spike (Supplementary Fig. 4a), which was proven correct experimentally by replacing the arginine with any other natural amino acid except for lysine (Figs. 3a, b, 4). The improved activity of optimized 19-77Δ mAbs was attributable to the loss of hydrogen bonding by R71 (Figs. 3c, 6c),

leading to greater flexibility of the CDRs as shown by molecular dynamics (Fig. 3e) and structural analyses (Fig. 6). This enhanced conformation flexibility, in turn, allowed each optimized antibody to be more tolerant of mutations within or near its epitope, ultimately resulting in the striking SARS-CoV-2 neutralization potency and breadth observed (Fig. 4). Remarkably, this optimization strategy was also successfully applied to 6 other IGHV3-53/3-66-derived human mAbs that target the same epitope cluster as 19-77 (Fig. 5). But its applicability did not extend to other SARS-CoV-2-neutralizing mAbs that utilize other VH germline genes, suggesting a specific mutation at framework residue 71 is not likely to confer a generalizable benefit to other mAbs. Nevertheless, the general concept of modifying antibody conformational flexibility should be further explored in other settings by other means. Increasing conformational rigidity could lead to improved antibody-antigen affinity, as has been reported for an IGHV4-59-encoded anti-lysozyme mAb[68]. On the other hand, increasing conformational flexibility, as shown in this study, could allow the antibody to better tolerate sequence variations in the target antigen. Such an approach may be useful for monoclonal antibodies directed to polymorphic antigens or surface proteins of rapidly evolving viruses such as HIV-1, coronaviruses, influenza viruses, and hepatitis C virus. It may be another important conceptual tool in our antibody engineering armamentarium.

## Methods

### Human participant

Blood sample from Donor 19, a 41-year-old Asian male, was collected at Columbia University Irving Medical Center. Donor 19 was confirmed for a BA.5 infection by PCR sequencing (single-nucleotide polymorphisms) and provided written informed consent. Sample collections were performed under protocols reviewed and approved by the Institutional Review Board of Columbia University. Clinical information of Donor 19 is provided in Supplementary Fig. 1a.

### Cell lines

Vero-E6 (CRL-1586) and HEK293T (CRL-3216) cells were purchased from the American Type Culture Collection (ATCC). Expi293 cells (A14527) were purchased from Thermo Fisher Scientific. Vero-E6-TMPRSS2-T2A-ACE2 (NR-54970) were obtained from BEI Resources. 293T-ACE2 were kindly provided by Dr. Jesse D. Bloom. The morphology of each cell line was confirmed visually before use. All cell lines tested mycoplasma negative. Vero-E6 and Vero-E6-TMPRSS2-T2A-ACE2 cell lines are from African green monkey kidneys. HEK293T, 293T-ACE2, and Expi293 cells are of female origin.

### Plasmid construction

SARS-CoV-2 spike-expressing plasmids for D614G, Alpha, Beta, Gamma, Delta and Omicron were previously generated[13,20,21,27]. Expressing constructs for the spike proteins of SARS-CoV-2-like sarbecoviruses were either previously generated[69] or newly generated by synthesizing (GenScript) spike genes and then cloned into the pCMV3 vector. To generate the expression constructs for soluble spike trimer (S2P) proteins, the ectodomains (1-1208aa, numbering based on WA1) of the spikes were PCR amplified and cloned into the paH vector and then introduced K986P and V987P substitutions, as well as a "GSAS" substitution of the furin cleavage site (682-685aa in WA1) into the spikes[70]. SARS-CoV S2P was fused with an AVI tag at the C terminus and D614G and XBB.1.5 S2P spikes were tagged with a 6×His tail also at the C terminus. To make the SARS-CoV-2 RBD-expressing construct, the RBD region (319–537aa) of each variant was fused with a 6×His tag and then cloned into the p3BNC vector. All constructs were confirmed by Sanger sequencing.

Antibody-expressing constructs were generated as previously described[38]. The variable regions of heavy and light chains for each antibody were synthesized (GenScript) and then cloned into the gWiz

vector. To make spike/antibody plasmid constructs carrying individual mutations, the Q5 Site-Directed Mutagenesis Kit (NEB) was utilized following the manufacturer's instructions.

## Protein purification

To make human ACE2-Fc (hACE2) protein, pcDNA3-sACE2-WT(732)-IgG1[71] (Addgene plasmid #154104, gift of Erik Procko) plasmid was transfected into Expi293 cells using 1 mg/mL polyethyleneimine (PEI) at a ratio of 1:3, and the supernatants were collected after 5 days. hACE2 was purified from the cell supernatant by using rProtein A Sepharose (GE). For antibody purification, both heavy and light chains of each antibody were transfected at a ratio of 1:1 into Expi293 cells using PEI. And the expressed antibody in the cell supernatant was purified using the same method as for hACE2 purification. For the spike trimer proteins or RBD proteins, paH-spike or p3BNC-RBD, respectively, was transfected into Expi293 cells using PEI at a ratio of 1:3, and the supernatants were collected five days later. The His-tagged and the AVI-tagged proteins were purified using Excel resin (Cytiva) and Agarose-bound Galanthus nivalis lectin (VectorLabs, AL-1243-5) according to the manufacturers' instructions. The molecular weight and purity were checked by running the proteins on SDS-PAGE.

After purification of 19-77 and 19-77△A/L/V, 100 μg of each antibody was prepared and run through a Superdex 200 Increase 10/300 GL column to generate their size exclusion chromatography (SEC) profiles. AVI-tagged SARS-CoV S2P protein were biotinylated using the BirA biotin-protein ligase standard reaction kit (Avidity LLC; BirA500) following the manufacturer's instructions.

## ELISA

50 ng per well of an antigen such as S2P spike, NTD (ACROBiosystems, S1D-C52H6), RBD (ACROBiosystems, SPD-C52H1), RBD-SD1 (Exonbio, 19Cov-S130), S1 (ACROBiosystems, S1N-C52H3), or S2 (ACROBiosystems, S2N-C52H2) was coated onto ELISA plates at 4 °C overnight. The ELISA plates were then blocked with 300 μL of blocking buffer consisting of phosphate-buffered saline (PBS) with 1% bovine serum albumin and 20% bovine calf serum (Sigma-Aldrich) at 37 °C for 2 h. Afterward, 100 μL of fivefold serially diluted antibodies was added and then incubated at 37 °C for 1 h. Next, 100 μL of 10,000-fold diluted Peroxidase AffiniPure goat anti-human IgG Fcγ fragment-specific antibody (Jackson ImmunoResearch, catalog no. 109-035-170, RRID: AB_2810887) was added into each well and incubated for another 1 h at 37 °C. The plates were washed between each step with PBST (0.5% Tween-20 in PBS). Last, 3,3′,5,5′-tetramethylbenzidine (TMB) substrate (Sigma-Aldrich) was added and incubated before the reaction was stopped using 1 M sulfuric acid. Absorbance was measured at 450 nm.

## Antibody pharmacokinetics in mice

About 100 μg (1 mg/mL) of each antibody was intraperitoneally (ip) injected into Balb/c mice (three mice for each antibody). Mouse blood was collected on days 2, 4, 7, and 10 post-IP injection and antibody concentrations in serum were measured by ELISA. Briefly, 100 ng goat anti-human IgG Fc antibody (Cat: 109-005-008, Jackson ImmunoResearch) was coated per well in 96-well plates overnight and the ELISA plates were then blocked with 300 μL of blocking buffer consisting of PBS with 1% bovine serum albumin and 20% bovine calf serum (Sigma-Aldrich) at 37 °C for 2 h. Afterward, 100 μL of threefold serially diluted serum or twofold serially diluted purified antibody was added and then incubated at 37 °C for 1 h. Next, 100 μL of 10,000-fold diluted Peroxidase AffiniPure goat anti-human IgG H + L HRP (Cat: 109-035-088, Jackson ImmunoResearch) was added into each well and incubated for another 1 hour at 37 °C. The plates were washed between each step with PBST (0.5% Tween-20 in PBS). Last, TMB substrate (Sigma-Aldrich) was added and incubated before the reaction was stopped using 1 M sulfuric acid. Absorbance was measured at 450 nm.

## Pseudovirus production and infectivity

SARS-CoV-2 pseudoviruses were generated as previously described[38]. In brief, HEK293T cells were transfected with a spike-expressing construct using 1 mg/mL PEI and then infected with VSV-G pseudotyped ΔG-luciferase (G*ΔG-luciferase, Kerafast) 1 day post-transfection. Two hours after infection, cells were washed three times with PBS, changed to fresh medium, and then cultured for one more day before the cell supernatants were harvested. Pseudoviruses in the cell supernatants were clarified by centrifugation, aliquoted, and stored at −80 °C.

To evaluate the infectivity of the pseudotyped escape variants of 19-77 and 19-77ΔV, fresh pseudoviruses without freezing and thawing were serially titrated from 50 μL with a dilution factor of 3 and then inoculated into Vero-E6 cells. After overnight culture, Vero-E6 cells were harvested and quantified for luciferase activity using the Luciferase Assay System (Promega).

## Pseudovirus neutralization assay

To normalize the viral input between assays before conducting the neutralization assays, pseudoviruses of SARS-CoV and SARS-CoV-2 variants and SARS-CoV-2-like sarbecoviruses were titrated on Vero-E6 cells and 293T-ACE2 cells. Heat-inactivated serum from Donor 19 was serially diluted starting from 1:25 with a dilution factor of four. Monoclonal antibodies were fivefold serially diluted starting from 20 μg/mL in 96-well plates in triplicate. Then, 50 μL of diluted pseudovirus was added and incubated with 50 μL serial dilutions of serum or mAb for 1 h at 37 °C. During the coculture, target cells were trypsinized, resuspended with fresh medium, and then added into the virus-sample mixture at a density of $4–10 \times 10^4$ cells/well. The plates were incubated at 37 °C for ~12 h before luciferase activity was quantified using the Luciferase Assay System (Promega) using SoftMax Pro v.7.0.2 (Molecular Devices). Neutralization $ID_{50}$ values for sera and $IC_{50}$ values for antibodies were calculated by fitting a nonlinear five-parameter dose-response curve to the data in GraphPad Prism v.10.

## Selection of escape mutations

SARS-CoV-2 isolate Omicron JN.1 (BEI NR-59693) was mixed with serial fivefold dilutions of 19-77ΔV antibody at an MOI of 0.2 and incubated for 1 h. Following incubation, the mix was overlaid on a 24-well plate bearing a monolayer of Vero-ACE2-TMPRSS2 cells (BEI NR-54970) to a final volume of 1 mL. Plates were incubated at 37 °C/ 5% $CO_2$ for 70 h till cytopathic effect (CPE) was complete (100%) in virus control wells bearing no antibody. At this time, all wells with antibody dilutions were scored to determine the 50% inhibition titer ($EC_{50}$) and supernatant collected from this well was used for the subsequent round of selection. Passaging of the progeny over new Vero-ACE2-TMPRSS2 cells continued till each of the virus variant was able to form CPE in the presence of 50 μg/mL of the antibody. The resulting supernatant was then collected, and RNA was extracted using QiaAMP Viral RNA kit (Qiagen 57704). cDNA was obtained using the Superscript IV enzyme (Thermo Scientific 18090010). Spike gene from the cDNA was amplified using limiting dilution nested PCR and sequenced using Sanger sequencing (Genewiz). Multiple clones from limiting dilution nested PCR were sequenced to confirm the dominant mutants in the pool of the resulting progeny viruses, and a percentage of their prevalence was calculated from the total number sequenced. At least eight clones were sequenced from each of the passages reported in Fig. 7.

To generate recombinant replication-competent VSV-ΔG bearing the JN.1 spike protein (VSV-ΔG-JN.1), the pVSVΔG-SARS-CoV-2-S_nLucP plasmid was purchased from Kerafast (Cat# EGA292) and its encoding SARS-CoV-2-S gene was replaced with the JN.1 spike gene to create pVSVΔG-JN.1_nLucP. The pVSVΔG-JN.1_nLucP plasmid and a set of helper plasmids, including VSV-N, VSV-P, VSV-L, and VSV-G (Kerafast, Cat# EH1012), were then transfected into 293T cells at a ratio of 5:3:5:1:8 using 1 mg/mL PEI-MAX. Before transfection, 293T cells were rinsed with serum-free DMEM, incubated with Vaccinia vTF7-3 (Imanis

Life Sciences, Cat# REA006) at an MOI of 5 for 45 min, and then replaced with fresh medium. Two days post-transfection, the supernatant was harvested and filtered through a 0.22 μm filter to remove cell debris and Vaccinia vTF7-3. The rVSV-ΔG-JN.1 generated from 293T cells was then serially diluted with a dilution factor of 5 and inoculated into Vero-E6-TMPRSS2-T2A-ACE2 cells in 24-well plates for 1 h. The virus was then washed away, and the cells were cultured at 37 °C/5% $CO_2$ for 16–24 h. Vero-E6-TMPRSS2-T2A-ACE2 cells were monitored, and the virus was harvested from wells in which only one plaque was observed, then filtered and stored at −80 °C. The rVSV-ΔG-JN.1 generated from Vero-E6-TMPRSS2-T2A-ACE2 cells was titrated on Vero-E6-TMPRSS2-T2A-ACE2 cells before use.

To select escape viruses, rVSVΔG-JN.1 was incubated with 0.4 μg/mL of 19-77ΔV for 1 h before being added to Vero-E6-TMPRSS2-T2A-ACE2 cells in 24-well plates at an MOI of 0.01. A total of 66 replicates were set up. Two days after coculture, cell supernatants from wells with CPE were harvested, and 100 μL of each supernatant was incubated with 4 μg/mL of 19-77ΔV for 1 hour before another round of infection in pre-seeded Vero-E6-TMPRSS2-T2A-ACE2 cells in 24-well plates. Two days later, supernatants containing escape viruses were further harvested and selected the same way using 20 μg/mL, and then 50 μg/mL of 19-77ΔV in a stepwise manner. mRNAs of the escape viruses in the supernatants were then extracted using the viral RNA/DNA purification kit (MACHEREY-NAGEL, Cat# 740643) and reverse-transcribed to cDNA using the SuperScript VILO cDNA Synthesis Kit (Invitrogen, Cat# 11754). The RBD genes of the escape viruses were then amplified using primers 5′-GGGCATCTACCAGACCAGCAACTTCA-3′ and 5′-GAACACATTGCTGCCTGTGCTGT-3′ and sequenced.

### Antigen-specific memory B cell sorting and single-cell B cell receptor sequencing
Peripheral blood mononuclear cells from Donor 19, and two healthy donors were stained with the LIVE/DEAD Fixable Yellow Dead Cell Stain Kit (Invitrogen) at ambient temperature for 20 min, followed by washing with RPMI 1640 complete medium [RPMI 1640 + 10% fetal bovine serum (FBS) + penicillin/streptomycin (P/S) (100 U/mL)] and incubation with 10 μg/mL XBB.1.5 S2P protein and biotinylated SARS-CoV S2P at 4 °C for 45 min. Afterwards, the cells were washed again and incubated with a cocktail of flow cytometry and Hashtag antibodies, consisting of CD3 PerCP-Cy5.5, CD19 APC/Cyanine 7, CD27 APC, IgM FITC, anti-His PE/DazzleTM 594, Streptavidin BV421, and human Hashtag 3 at 4 °C for 1 h. Stained cells were then washed, resuspended in RPMI 1640 complete medium, and sorted for SARS-CoV and/or XBB.1.5 S2P trimer-specific memory B cells (CD3 − CD19 + CD27+IgM −antigen+ live single lymphocytes) by flow cytometry. The sorted cells were mixed with spike-in CD3+ cells and loaded into a 10X Chromium chip of the 5′ Single Cell Immune Profiling Assay (10X Genomics) at the Columbia University Single-Cell Analysis Core. Library preparation and quality control were performed according to the manufacturer's protocol and sequenced on a NextSeq 500 sequencer (Illumina).

### Identification of spike-specific antibody transcripts
Antibody transcripts specific to the XBB.1.5 S2P spike and SARS-CoV spike trimers were identified following previously established methods[38]. The assembly of full-length antibody transcripts was performed utilizing the Cell Ranger V(D)J analysis software (version 3.1.0, 10X Genomics), employing default settings with the GRCh38 V(D)J germline sequence version 2.0.0 as the reference genome. To differentiate between cells captured in the antigen-specific sorting process and those added as spike-ins, the count module of Cell Ranger was used to quantify the presence of all hashtag oligonucleotides in each cell based on Next Generation Sequencing (NGS) raw data. Identification of high-confidence antigen-specific cells was achieved using the following criteria: (1) A minimum of 100 copies of the antigen-specific

hashtag was required for a cell to be classified as antigen-specific, (2) Given that hashtags might detach from their original cells and attach to others within the sample, a cell was considered truly antigen-specific only if the copy number of its specific hashtag was at least 1.5 times higher than that of any non-specific hashtag present, (3) Cells deemed to be of low quality were excluded based on the cell quality assessment algorithm used by Cell Ranger, (4) Only cells expressing both productive heavy and light chain antibody gene pairs were retained, (5) In instances where a cell exhibited more than two transcripts for heavy and/or light chains, transcripts supported by fewer than three unique molecular identifiers (UMIs) were discarded, and 6) Cells sharing identical heavy and light chain sequences, potentially indicative of mRNA contamination, were consolidated into a single cell entry.

### Antibody transcript annotation
Transcripts specific to the antigen were analyzed and annotated with SONAR version 2.0, following previously established procedures. Assignment of V(D)J gene segments to each transcript was conducted via BLASTn, employing specialized parameters against a germline gene repository sourced from the International ImMunoGeneTics (IMGT) information system database. The identification of the complementarity-determining region 3 (CDR3) utilized BLAST alignments of the V and J segments, focusing on the conserved second cysteine within the V segment and the WGXG (for heavy chains) or FGXG (for light chains) motifs in the J segment, with "X" indicating any amino acid. Isotype determination for heavy chain transcripts was achieved by analyzing Constant domain 1 (CH1) sequences against a human CH1 gene database from IMGT, using BLASTn with standard parameters. The CH1 allele presenting the lowest E-value was selected for precise isotype classification, adhering to a BLAST E-value cutoff of 10e-6. Transcripts with incomplete V(D)J segments, frameshifts, or extraneous sequences beyond the V(D)J region were discarded. The filtered transcripts were then aligned to their corresponding germline V gene using CLUSTALO, and levels of somatic hypermutation were quantified through the Sievers method. In instances where cells possessed multiple high-quality heavy or light chains, potentially indicative of doublets, combinations of all H and L chains were generated.

### Crystallization and data processing
19-77 Fab was produced by digestion of IgG with immobilized Endoproteinase Lys-C (Sigma-Aldrich) equilibrated with 25 mM Tris pH 8.5 and 1 mM EDTA for 3 h. The resulting Fab was further purified from the cleaved Fc domain by cation exchange chromatography. Fab purity was analyzed by SDS-PAGE and buffer-exchanged into 20 mM Tris, 150 mM, pH 7.4 prior to cryo-EM/Crystallization experiments.

19-77/SARS-CoV-2-RBD and 19-77/EG5.1-CoV-2-RBD complexes were prepared by mixing each of the protein components at an equimolar concentration and incubating overnight at 4 °C. Protein complexes were then isolated by gel filtration on a Superdex 200 column (Cytiva, GE Healthcare). Fractions containing complexes were pooled and concentrated to 12.0 mg/mL in SEC buffer. Screening for initial crystallization conditions was carried out in 96-well sitting drop plates using the vapor-diffusion method with a mosquito crystallization robot (TTP LabTech) using various commercially available crystallization screens: MSCG-1 (Anthracene), Proplex and LMB (Molecular dimensions). Diffraction quality crystals were obtained after 7 days in the following conditions for 19-77/SARS-CoV-2-RBD: 0.1 M NaCl, 0.1 M Tris pH 7.5, 12% w/v PEG 4000, and the following conditions for 19-77/EG5.1-CoV-2-RBD: 75% MPD and 0.1 M HEPES pH 7.5.

Prior to data collection, crystals were cryoprotected with 40% ethylene glycol supplemented in mother liquor and flash frozen in liquid nitrogen. X-ray diffraction data extending to 2.8 Å (19-77/SARS-CoV-2-RBD) and 3.2 Å (19-77/ EG5.1-CoV-2-RBD) resolution were collected at 100 K on beam line 17-ID-1 (AMX) at Brookhaven National Laboratory. Diffraction data were processed with XDS[72] and scaled

using AIMLESS[73] from the CCP4 software suite (Collaborative Computational Project Number 4, 1994)[74]. Molecular replacement was performed with PHASER[75], using a previously reported RBD structure (PDB 7L5B) and for 19-77 Fab, heavy chain (PDB 7XIK), light chain (3FIK) used as search models. Manual rebuilding of the structure using COOT[76] was alternated with refinement using Phenix refine[77]. The Molprobity server was used for structure validation[78] and PyMOL (version 2.1, Schrödinger, LLC) for structure visualization. A summary of the X-ray data collection and refinement statistics are shown in Supplementary Fig. 3c.

## Cryo-EM sample preparation
Fab fragments of antibodies were produced by digestion of IgG with immobilized Endoproteinase Lys-C (Sigma-Aldrich) equilibrated with 25 mM Tris, pH 8.5, and 1 mM EDTA for 3 h. The resulting Fabs were purified by ion-exchange chromatography on a mono-Q column.

For the structure of 19-77 bound to D614G spike, the complex was made by mixing purified SARS-CoV-2 S2P D614G spike protein with Fab in a 1:3 molar ratio (spike potomer:Fab) in PBS, pH 7.4, such that the final concentration of spike was 1 mg/mL. This mixture was incubated on ice for 1 h.

For the structures of the ternary complex of 19-77, RBDs, and S309, complexes were made by mixing purified SARS-CoV-2 RBD with Fabs in a 1:1.2 molar ratio in PBS, pH 7.4, and further purified using size exclusion chromatography (SEC) using a Superdex 200 Increase column. The resulting complex peak was then concentrated to 4 mg/mL and held on ice until vitrification.

Before freezing, 0.005% (w/v) n-dodecyl β-D-maltoside (DDM) was added to deter preferred orientation and aggregation during vitrification. Cryo-EM grids were prepared by applying 3 μL of sample to a freshly glow-discharged carbon-coated copper grid (CF 1.2/1.3 300 mesh); the sample was then vitrified in liquid ethane using a Vitrobot Mark IV with a wait time of 30 s, a blot time of 3 s, and a blot force of 0.

## Cryo-EM data collection and analysis
Cryo-EM data for single particle analysis were collected at the Columbia Cryo-EM Facility on a Titan Krios electron microscope operating at 300 kV, equipped with a Gatan K3-BioQuantum detector and energy filter, using the Leginon7 software package. Exposures were taken at a magnification of 105,000x (pixel size of 0.83 Å), using a total electron flux of 58 e-/Å2 fractionated over 50 frames with an exposure time of 2.5 s. A random defocus range of −0.8 to −2.0 μm was used.

Data processing was performed using cryoSPARC v3.3.1.8. Raw movies were aligned and dose-weighted using patch motion correction, and the micrograph contrast transfer function (CTF) parameters were estimated using patch CTF estimation. Micrographs were picked using a blob or template picker, and an initial particle set was selected using 2D classification. Further heterogeneous refinement was used to 3D-classify particles and remove debris. The resulting curated particle sets were corrected for local motion and refined using homogenous refinement. Local refinement was performed for the spike dataset using a mask enveloping the RBD+Fab variable region. The default cryoSPARC auto-sharpened maps were then used to build the models. Cryo-EM data collection and consensus refinements are summarized in Supplementary Fig. 3.

## Model building and refinement
Initial molecular models for Fabs were generated using Alphafold Multimer[79] using paired heavy and light sequences. An RBD from PDB 7KNI, an RBD-up spike structure, was used as a starting model. For the initial 19-77 structure, the 14-7 structure (PDB 8F89), RBD (PDB 8IOV), and S309 Fab (PDB 7XSW) were used. The initial models were rigid body docked into the density map using UCSF Chimera's "fit to map" tool and combined. The Fab CDR loops were manually fitted to the density map using Coot real-space refinement. The models were fit to density using the ISOLDE package in ChimeraX[80]. Ramachandran outliers were corrected using ISOLDE's "flip peptide bond" feature. Real-space refinement in Phenix[81] was performed to remove geometry outliers. The remaining manual adjustments were performed in Coot. Models were validated using MolProbity16 in Phenix and the PDB validation server and deposited to the PDB with accession codes: 19-77 + SARS-CoV-2 D614G RBD (PDB 9CFE), 19-77 + HK.3 RBD (PDB 9CFF), 19-77ΔV + HK.3 RBD (PDB 9CFG), and 19-77ΔV + JD.1.1 RBD (PDB 9CFH). A summary of data collection, processing, and model refinement statistics is shown in Supplementary Fig. 3.

## In silico antibody engineering
The energy changes for the binding of both the mutant and original 19-77 to the HK.3 RBD were calculated using FoldX software[40]. Initially, the 19-77 and HK.3 complex was repaired and optimized using the "Optimize" function. Subsequently, every position in the heavy and light chains was subjected to saturation mutagenesis to all other amino acids using the "BuildModel" function. This process generated both an unmutated model and a mutant model for each mutation. The binding energies were estimated with the "AnalyseComplex" function, and the changes in Gibbs free energy ($\Delta\Delta G$) were calculated based on the energy difference between each mutant and its corresponding original antibody-RBD complex.

## Molecular dynamics analysis
Antibody 19-77, along with other VH3-53/66 class antibodies, was meticulously aligned through the RBD within each complex to ensure structural consistency. The R71A mutant antibodies were precisely engineered using the "Mutagenesis" function in PyMOL version 2.5.4, provided by Schrödinger, LLC. Subsequently, the Fab regions of both the original and mutant antibodies underwent molecular dynamics simulations employing GROMACS on the WebGro server. The preprocessing was executed using the GROMOS96 54a7 force field and incorporated an SPC water model within a cubic simulation box, supplemented with 0.15 M NaCl to mimic physiological ionic strength. The initial energy minimization step was set to 5000 iterations to stabilize the system. This was followed by equilibration and simulation phases under NVT/NPT conditions, conducted at a physiological temperature of 310 K for 10 ns, with all other parameters set to default. To ensure the reliability of the simulation, the root mean square deviation (RMSD) of each system was manually monitored to verify that each simulation reached equilibrium. Additionally, the per-residue root-mean-square fluctuation (RMSF) analysis was conducted to assess the flexibility of residues within each Fab region.

## Structure comparison analysis
The paratope and epitope residues for 19-77 was identified using PISA with the default parameters. The gene-specific substitution profiles (GSSP) for 19-77 germline genes were obtained from the cAb-Rep database (https://cab-rep.c2b2.columbia.edu/). The 19-77 and its mutant complexes were first superimposed by using the "align" function in PyMOL 2.5.4. The distance between the Cα from identical residues within the 19-77 heavy chains were then determined using the rms_cur function in PyMOL by in-house Python script.

## Data analysis
$IC_{50}$ values were determined by fitting the data to five-parameter dose-response curves in GraphPad Prism v.10.3. GraphPad Prism v.10.3 was also used for data visualization. PISA was used for identifying antibody-spike interface residuals. PyMOL V.2.5.4 was used to perform mutagenesis and to generate structural plots.

**Reporting summary**

Further information on research design is available in the Nature Portfolio Reporting Summary linked to this article.

## Data availability

All experimental data were provided in this manuscript. SARS-CoV-2 spike sequences were sourced from the Global Initiative on Sharing All Influenza Data (GISAID) database (https://www.gisaid.org/). The sequences for 19-77 and 19-79 are deposited in NCBI under accession: PV010127 to PV010130 [https://www.ncbi.nlm.nih.gov/nuccore/PV010127, https://www.ncbi.nlm.nih.gov/nuccore/PV010128, https://www.ncbi.nlm.nih.gov/nuccore/PV010129, https://www.ncbi.nlm.nih.gov/nuccore/PV010130]. Structures are available in the Protein Data Bank (PDB) under the following IDs: 9CFE, 9CFF, 9CFG, 9CFH, and 9CAV. All data supporting the results of this study are available in the article, supplementary, and source data files or from the corresponding author upon request. Source data are provided with this paper.

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

## Acknowledgements

This study was supported by funding from the Gates Foundation (INV019355) and donations from Andrew and Peggy Cherng.

## Author contributions

D.D.H. and L.L. conceived this project. J.Y., J.H., and C.C.T. constructed antibody expression plasmids. R.G.C. conducted cryo-EM and E.R.R. conducted x-ray. J.Y. studied antibody PK in mice. M.S.N. and Y.H. performed escape virus selection on live virus. Y.G. did bioinformatic and structural analyses. Q.W., I.A.M., M.W., H.H., C.C.T. and L.L. conducted pseudovirus neutralization assays. Q.W. and L.L. conducted all other experiments including antigen-specific B cell sorting, protein expression plasmid construction, ELISA, protein/antibody purification, rVSV-JN.1 escape variants selection, etc. L.S., L.L., and D.D.H. directed and supervised the project. Q.W., Y.G., R.G.C., L.L., L.S., and D.D.H. analyzed the results and wrote the manuscript.

## Competing interests

Q.W., J.Y., R.G.C., Y.G., L.S., L.L., and D.D.H. are inventors on the patent application (WH Ref: 0019240.01338US1) or the provisional patent application (63/619,716) filed by Columbia University for a number of SARS-CoV-2-neutralizing antibodies described in this manuscript. Both sets of applications are under review. D.D.H. is a co-founder of TaiMed Biologics and RenBio, a consultant to WuXi Biologics and Brii Biosciences, and board director for Vicarious Surgical. The remaining authors declare no competing interests.
