## [Transparent Peer Review file · Nature Communications]

Optimizing a Human Monoclonal Antibody for Better Neutralization of SARS-CoV-2

Corresponding Author: Dr David Ho

Version 0:

Reviewer comments:

Reviewer #1

(Remarks to the Author)

The authors report engineering of 19-77 (an anti-Spike RBD mAb which suffered full resistance from the common A475V mutation) into the 19-77Delta mAbs harboring R71A, R71L, and R71V. Engineering was approached not by focusing on increasing affinity, but rather on increasing conformational flexibility. These mutations were proven helpful for other class 1 anti-Spike mAbs within the VH3-53/66 family (likely via increased CDR3 flexibility), which could have terrific translational potential. The Delta mAbs remained resistant to F456S, Y473S, A475D, G476D, N487H, and Y489H, which currently remain rare mutations. The manuscript is very well written and comprehensive, but I would suggest a better discussion on the risks of antibody monotherapies when used for treatment as opposed to pre-exposure prophylaxis (see, e.g. PMID 39630849 and 38735657).

Line 31: change "This evolution has rendered inactive all therapeutic monoclonal antibodies previously authorized, and it is now 31 threatening the remaining clinical product for immunoprophylaxis against COVID-19" into "This evolution has rendered inactive all anti-Spike therapeutic monoclonal antibodies previously authorized, and it is now threatening the remaining pipeline".

Shouldn't protein names be capitalized ? E.g. Spike

It would be fundamental to assess 19-77Delta's efficacy against LP.8.1* sublineage, which is ramping over XEC these days.

Line 62 : change "gain" into "gained"

Line 80 : change "BA.2.86 became JN.1 that has since gained dominance worldwide from late 2023 until recently" into "BA.2.86 became JN.1, whose progenitors have since gained dominance worldwide from late 2023 until recently"

Line 83: change "In March of this year" into "In March 2024". "Permagard" should be "Pemgarda", but since it is a brand name it should be removed. Please also quote AstraZeneca's sipavibart (ADZ3152) within the pipeline, since their RCT is going to be published in days.

Line 88: change "Currently, the dominant SARS-CoV-2 is KP.3.1.1, and the emergent XEC is steadily gaining ground" into "Currently, the dominant SARS-CoV-2 sublineages are KP.3.1.1 and XEC"

Line 90: please also cite "Yao T, Ma Z, Lan K, et al. Neutralizing Activity and Viral Escape of Pemivibart by SARS-CoV-2 JN.1 sublineages. bioRxiv. 2024:2024.11.08.622746" and "Wang Q, Guo Y, Mellis IA, et al. Antibody evasiveness of SARS-CoV-2 subvariants KP.3.1.1 and XEC. bioRxiv. 2024:2024.11.17.624037" in addition to the self-citation.

Line 166: change "HK.3 (XBB.1.5 carrying Q52H and L455F/F456L mutations)" into "HK.3 (XBB.1.9.2 carrying Q52H and L455F/F456L mutations)". Change "JF.1 (XBB.1.5 carrying E180V, L455F/F456L, and K478R)" into "JF.1 (XBB.1.16.6 carrying E180V, L455F/F456L, and K478R)"

Line 292: was F456S caused by a single nucleotide (TTT to TCT) or a 2-nucleotide (TTT to TCC/TCA/TCG) mutation? This is intriguing to understand chances of immune escape.

Line 304 : "were detected in in 11 wells" remove duplicate

Line 311: again , was G485D a single (GGT to GAT) or 2-nucleotide mutation (GGT to GAC/CGT) ?

Line 580 : change "SARS-Cov-2" into "SARS-CoV-2"

Reviewer #2

(Remarks to the Author)

The authors report the isolation, characterization, and optimization of a novel neutralizing monoclonal antibody (mAb 19-77) targeting the SARS-CoV-2 spike protein. They find that while the original isolated mAb 19-77 exhibits strong neutralization

potency and breadth against many SARS-CoV-2 variants, it fails to neutralize the HK.3 and JF.1 variants. Cryo-EM analysis revealed that 19-77 binds to an epitope on the receptor-binding domain (RBD) that overlaps with the ACE2 binding motif. Through molecular dynamics simulations and *in silico* mutagenesis, the authors identified residue R71 in the heavy chain framework as a key determinant of antibody flexibility, specifically influencing the conformations of CDRH1 and CDRH2. Experimental validation showed that mutating R71, most notably to alanine (R71A), enhanced both the potency and breadth of 19-77, effectively restoring its neutralization capability against HK.3 and JF.1. Moreover, analogous R71 mutations in other VH3-53/66 class antibodies were found to similarly improve their neutralization breadth. The study also identified and characterized potential viral escape variants; however, these mutations are very rare in nature, likely due to an associated reduction in viral fitness. Overall, this robust and well-planned investigation provides valuable insights into antibody optimization strategies, with implications for enhancing therapeutic efficacy against emerging SARS-CoV-2 variants. The breadth of structural work is impressive. However, thorough description and details of image processing, especially for single particle cryoEM are missing. Supplementary figure 2 does not provide sufficient detail to understand the protocol used for data collection. I would like to see a series of supplementary figures describing the image processing workflow for each of the cryoEM experiments, including the number of micrographs used, initial particles selected, method of selection, representative 2D class averages, 3D class averages, etc. In summary, I believe that this manuscript is important and suitable for publication with the requested additions.

Reviewer #3

(Remarks to the Author)

Remarks to the Author:

In this manuscript Wang, Guo and Casner et al. describe the isolation and characterization of a broad but incomplete SARS-CoV-2 neutralizing antibody named 19-77 derived from the known germline IGHV3-53. Through structural analyses and mutational scanning, the authors have identified a specific residue to be mutated in the framework region 3 of the heavy chain (R71) that increased antibody flexibility improving the antibody potency and breadth. The mutated version of this antibody was named 19-77 Δ . Interestingly, the increased potency and breadth demonstrated for 19-77 Δ was also shown for other 5 mAbs deriving from the IGHV3-53 and closely related 3-66 germlines, suggesting that mutations at position R71 can benefit this class of antibodies and that greater conformational flexibility of CDRs can improve antibody functionalities. Subsequently the authors performed experiments to select *in vitro* SARS-CoV-2 resistant variants for 19-77 Δ and found different mutations that could completely abrogate the neutralization activity of all 19-77 Δ optimized forms. Worth noting, the escape mutations only rarely appeared in nature.

Overall, this is a well thought and experimentally executed work which highlights the importance of antibody engineering to provide novel and long-standing medical tools to patients in need. I suggest the acceptance of this work for publication in Nature Communications.

Minor comments:

- *In vitro* should be written in italics throughout the text.
- Line 184, *in silico* should be written in italics.
- Line 129, are "data not shown" allowed?
- Line 242, can they refer to "our unpublished mAb"?
- I would probably incorporate Figure 3 and 4 or 4 and 5.

Version 1:

Reviewer comments:

Reviewer #1

(Remarks to the Author)

All my concerns have been successfully addressed in the revised version

Reviewer #2

(Remarks to the Author)

The authors have addressed my comments in a satisfactory manner.

To the Reviewers,

We appreciate the reviewers' insightful and positive comments, which have helped us improve our manuscript. In response to their feedback and requested revisions, we have incorporated additional experimental data and revised the text accordingly, with all changes highlighted using 'Track Changes' in the Word document. Relevant raw data have also been included in the Source Data File. Below, we provide our responses to each of the reviewers' points (in Arial font), with corresponding changes indicated in cyan.

Reviewer #1 (Remarks to the Author):

The authors report engineering of 19-77 (an anti-Spike RBD mAb which suffered full resistance from the common A475V mutation) into the 19-77Delta mAbs harboring R71A, R71L, and R71V. Engineering was approached not by focusing on increasing affinity, but rather on increasing conformational flexibility. These mutations were proven helpful for other class 1 anti-Spike mAbs within the VH3-53/66 family (likely via increased CDR3 flexibility), which could have terrific translational potential. The Delta mAbs remained resistant to F456S, Y473S, A475D, G476D, N487H, and Y489H, which currently remain rare mutations. The manuscript is very well written and comprehensive, but I would suggest a better discussion on the risks of antibody monotherapies when used for treatment as opposed to pre-exposure prophylaxis (see, e.g. PMID 39630849 and 38735657).

Response: We agree with the comment regarding the risks associated with antibody monotherapies and have added clarifying statements to the Discussion section to highlight this point for readers.

Line 31: change "This evolution has rendered inactive all therapeutic monoclonal antibodies previously authorized, and it is now 31 threatening the remaining clinical product for immunoprophylaxis against COVID-19" into "This evolution has rendered inactive all anti-Spike therapeutic monoclonal antibodies previously authorized, and it is now threatening the remainig pipeline".

Response: Revised. Thank you.

Shouldn't protein names be capitalized ? E.g. Spike

Response: Thank you for your comments, the protein names should be capitalized. However, in the COVID field, most publications use spike, not Spike.

It would be fundamental to assess 19-77Delta's efficacy against LP.8.1* sublineage, which is ramping over XEC these days.

Response: Thank you for your valuable comments, we have added variants like LP.8.1, LP.8.1.1, MC.10.1, and LF.7 in the revised figure 4. The results showed that 19-77_R71V could still potentially neutralize these recent variants.

Line 62 : change "gain" into "gained"

Response: Revised.

Line 80 : change "BA.2.86 became JN.1 that has since gained dominance worldwide from late 2023 until recently" into "BA.2.86 became JN.1, whose progenitors have since gained dominance worldwide from late 2023 until recently"

Response: We have updated the text to read "...BA.2.86 became JN.1, and the JN.1 sublineage has been dominant worldwide from late 2023 until recently."

Line 83: change "In March of this year" into "In March 2024". "Permagard" should be "Pemgarda", but since it is a brand name it should be removed. Please also quote AstraZeneca's sipavibart (ADZ3152) within the pipeline, since their RCT is going to be published in days.

Response: Thank you for your valuable comments. We have added the information for AZD3152 in the revised manuscript. This antibody has reduced potency against XBB subvariants containing F456L, the current dominant JN.1 variants all carry F456L, such as XEC and LP.8.1.

Line 88: change "Currently, the dominant SARS-CoV-2 is KP.3.1.1, and the emergent XEC is steadily gaining ground" into "Currently, the dominant SARS-CoV-2 sublineages are KP.3.1.1 and XEC"

Response: Revised.

Line 90: please also cite "Yao T, Ma Z, Lan K, et al. Neutralizing Activity and Viral Escape of Pemivibart by SARS-CoV-2 JN.1 sublineages. bioRxiv. 2024:2024.11.08.622746" and "Wang Q, Guo Y, Mellis IA, et al. Antibody evasiveness of SARS-CoV-2 subvariants KP.3.1.1 and XEC. bioRxiv. 2024:2024.11.17.624037" in addition to the self-citation.

Response: Revised.

Line 166: change "HK.3 (XBB.1.5 carrying Q52H and L455F/F456L mutations)" into "HK.3 (XBB.1.9.2 carrying Q52H and L455F/F456L mutations)". Change "JF.1 (XBB.1.5 carrying E180V, L455F/F456L, and K478R)" into "JF.1 (XBB.1.16.6 carrying E180V, L455F/F456L, and K478R)"

Response: Revised.

Line 292: was F456S caused by a single nucleotide (TTT to TCT) or a 2-nucleotide

(TTT to TCC/TCA/TCG) mutation? This is intriguing to understand chances of immune escape.

Response: Thank you for your valuable comments. We have added the nucleotide changes for the example mutations you suggested and included a sentence stating that all mutations selected in our escape experiments resulted solely from single nucleotide changes in line 326 and 346.

Line 304 : "were detected in in 11 wells" remove duplicate

Response: Revised.

Line 311: again , was G485D a single (GGT to GAT) or 2-nucleotide mutation (GGT to GAC/CGT) ?

Response: See previous comments.

Line 580 : change "SARS-Cov-2" into "SARS-CoV-2"

Response: Revised.

Reviewer #2 (Remarks to the Author):

The authors report the isolation, characterization, and optimization of a novel neutralizing monoclonal antibody (mAb 19-77) targeting the SARS-CoV-2 spike protein. They find that while the original isolated mAb 19-77 exhibits strong neutralization potency and breadth against many SARS-CoV-2 variants, it fails to neutralize the HK.3 and JF.1 variants. Cryo-EM analysis revealed that 19-77 binds to an epitope on the receptor-binding domain (RBD) that overlaps with the ACE2 binding motif. Through molecular dynamics simulations and in silico mutagenesis, the authors identified residue R71 in the heavy chain framework as a key determinant of antibody flexibility, specifically influencing the conformations of CDRH1 and CDRH2. Experimental validation showed that mutating R71, most notably to alanine (R71A), enhanced both the potency and breadth of 19-77, effectively restoring its neutralization capability against HK.3 and JF.1. Moreover, analogous R71 mutations in other VH3-53/66 class antibodies were found to similarly improve their neutralization breadth. The study also identified and characterized potential viral escape variants; however, these mutations are very rare in nature, likely due to an associated reduction in viral fitness. Overall, this robust and well-planned investigation provides valuable insights into antibody optimization strategies, with implications for enhancing therapeutic efficacy against emerging SARS-CoV-2 variants.

Response: Thank you for your positive comments.

The breadth of structural work is impressive. However, thorough description and details

of image processing, especially for single particle cryoEM are missing. Supplementary figure 2 does not provide sufficient detail to understand the protocol used for data collection. I would like to see a series of supplementary figures describing the image processing workflow for each of the cryoEM experiments, including the number of micrographs used, initial particles selected, method of selection, representative 2D class averages, 3D class averages, etc.

Response: We appreciate the reviewer's valuable comments. Accordingly, we have added a new Extended Data Fig. 2, which provides a detailed description of the image processing workflow, particularly for single-particle cryo-EM analysis.

In summary, I believe that this manuscript is important and suitable for publication with the requested additions.

Reviewer #3 (Remarks to the Author):

Remarks to the Author:

In this manuscript Wang, Guo and Casner et al. describe the isolation and characterization of a broad but incomplete SARS-CoV-2 neutralizing antibody named 19-77 derived from the known germline IGHV3-53. Through structural analyses and mutational scanning, the authors have identified a specific residue to be mutated in the framework region 3 of the heavy chain (R71) that increased antibody flexibility improving the antibody potency and breadth. The mutated version of this antibody was named 19-77 Δ . Interestingly, the increased potency and breadth demonstrated for 19-77 Δ was also shown for other 5 mAbs deriving from the IGHV3-53 and closely related 3-66 germlines, suggesting that mutations at position R71 can benefit this class of antibodies and that greater conformational flexibility of CDRs can improve antibody functionalities. Subsequently the authors performed experiments to select in vitro SARS-CoV-2 resistant variants for 19-77 Δ and found different mutations that could completely abrogate the neutralization activity of all 19-77 Δ optimized forms. Worth noting, the escape mutations only rarely appeared in nature.

Overall, this is a well thought and experimentally executed work which highlights the importance of antibody engineering to provide novel and long-standing medical tools to patients in need. I suggest the acceptance of this work for publication in Nature Communications.

Response: Thank you for your valuable comments on our manuscript.

Minor comments:

- In vitro should be written in italics throughout the text.

Response: Revised.

- Line 184, *in silico* should be written in italics.

Response: Revised.

- Line 129, are “data not shown” allowed?

Response: Revised.

- Line 242, can they refer to “our unpublished mAb”?

Response: We have added the information of 19-79 mAb into this manuscript and deposited its sequence to NCBI under accession PV010129 and PV010130.

- I would probably incorporate Figure 3 and 4 or 4 and 5.

Response: Thank you for your comments. We believe the current arrangement of figures is appropriate, as these three figures address distinct topics and should therefore be considered independently.

REVIEWERS' COMMENTS

Reviewer #1 (Remarks to the Author):

All my concerns have been successfully addressed in the revised version

Response: Thank you very much for your thoughtful review and positive feedback. We are glad to hear that our revisions addressed your concerns satisfactorily.

Reviewer #2 (Remarks to the Author):

The authors have addressed my comments in a satisfactory manner.

Response: Thank you for your helpful comments and for acknowledging our revisions. We appreciate your constructive input throughout the review process.